# Inequalities and determinants of unmet need for SARS-CoV-2 testing in Ghana, Burkina Faso and Madagascar (2020 – 2021)

Jacob Novignon[1,13], John H. Amuasi[2,3,4,13], Eva Lorenz[4,5], Thierry A. Ouedraogo[6], Valentin Boudo[6], Boubacar Coulibaly[6,7], Ali Sié[6,7,8], Anthony Afum-Adjei Awuah[2,9], Daniela Fusco[5,10], Rivo A. Rakotoarivelo[11], Jürgen May[4,5,12], Aurélia Souares[7,8], Nicole S. Struck [4,5,14] ✉ & Manuela De Allegri[8,14]

## Abstract

**Background** Many sub-Saharan African countries faced considerable challenges in containing SARS-COV-2 transmission due to limited capacity to test and treat all those affected. While previous studies have measured testing uptake, no study has examined uptake in relation to WHO testing criteria. We assess unmet need for SARS-CoV-2 testing in Ghana, Burkina Faso, and Madagascar as a measure of health system preparedness, defined as the gap between individuals meeting WHO testing criteria and those actually tested.
**Methods** We used urban population-based data from 2434 households in Ghana, Burkina Faso, and Madagascar sampled between February and May 2021. We defined unmet need for testing using three measures (serological, exposure-based, symptom-based) aligned with WHO testing criteria and used logistic regression models to examine associated factors and concentration indices to quantify socioeconomic inequalities.
**Results** Here we show that unmet need for SARS-CoV-2 testing is high, ranging from 93% to 95% across all three countries, regardless of the measure used. Socioeconomic inequalities exist, with the poorest quintile experiencing the highest unmet need. Wealth, travel history outside the city, and perceived COVID-19 risk are associated with lower unmet need.
**Conclusions** Our findings reveal critical weaknesses in health system preparedness that call for urgent measures to strengthen testing capacity and ensure equitable access for both routine care provision and future health crises.

## Plain language summary

Studies during the COVID-19 pandemic showed that far more people in sub-Saharan Africa had been exposed to the virus than official numbers suggested. This gap occurred because testing capacity was severely limited. We measured "unmet need for testing" by comparing people who should have been tested according to World Health Organization guidelines with those who actually received tests. Using data from 2,434 urban households in Ghana, Burkina Faso, and Madagascar collected in 2021, we compared people who were tested against people with symptoms, disease exposure, or evidence of past infection. Over 90% of people who should have been tested were not. Poor people were least likely to receive tests, while wealthier individuals had better access. These findings reveal critical healthcare inequalities. Strengthening testing systems and ensuring fair access for everyone, especially vulnerable populations, is essential for managing both current healthcare needs and future health emergencies.

In 2005, the 58th World Health Assembly adopted the International Health Regulations (IHR), a legal framework endorsed by 196 countries defining national rights and obligations during public health emergencies of international concern. The IHR compels countries to i) detect potential threats, ii) report them to the World Health Organization (WHO), iii) coordinate responses with other countries, and iv) respond effectively[1]. Fulfilling these functions depends on health system preparedness, defined as the capacity to anticipate and respond to population needs, even during crisis[2], encompassing prevention, protection, and effective public health response.

The COVID-19 pandemic exposed weaknesses in health system preparedness, with shortcomings evident across all areas of pandemic management, from prevention and detection to reporting and treatment[3]. Gaps

were particularly pronounced in low-income settings, where limited capacities and underfunded infrastructure impaired effective response[4]. While officially reported COVID-19 deaths in sub-Saharan Africa (SSA) were lower than in other regions[5], the continent faced severe indirect impacts, including disrupted malaria control efforts that reversed 20 years of progress[6].

Burkina Faso, Ghana, and Madagascar reported their first COVID-19 cases in March 2020, with official statistics documenting 20,729, 62,844, and 158,159 confirmed cases, respectively, by February 2021[5]. Many SSA countries struggled to contain SARS-COV-2 transmission with insufficient testing and treatment capacity[3,4]. By October 2021, only 1.8% of the 4 billion COVID-19 tests performed globally were conducted in Africa[7]. In 2020, testing rates in selected African countries ranged from 2.18 per thousand people in Ethiopia to 41.68 in South Africa, compared to Western countries: 118.58 in the UK, 103.57 in Italy, 87.13 in Spain, and 138.17 in USA[8]. Only symptomatic individuals and international travelers were typically tested, with no organized mass testing campaigns[9]. These constraints in case detection likely contributed to substantial underreporting. Seroprevalence studies in South Sudan and Zambia revealed that actual infections were 100 and 92 times higher, respectively, than officially reported cases[10,11].

Accra, the capital of Ghana, with approximately 2 million residents, reported more than 50% of the country's confirmed cases[12]. Kumasi, Ghana's second-largest city with a population of 1.5 million and a major commercial hub, was identified as another hotspot[12,13].

In Burkina Faso, Ouagadougou (the capital and administrative center) and Bobo-Dioulasso (the economic capital) represented the country's two largest urban centers with the highest reported case concentrations[13]. In Madagascar, Fianarantsoa, the country's fourth-largest city, located in the central highlands, represented regional transmission patterns outside the capital Antananarivo. These urban centers across the three countries were critical for understanding the pandemic's true extent across diverse geographic contexts in SSA[13].

Regardless of debates about COVID-19 statistics accuracy, real-time testing capacity was clearly insufficient across many SSA countries[14]. As a fundamental component of health system preparedness, diagnostic services allow for the distinction of the healthy from sick individuals, creating opportunities for prevention, and opening ways to treatment for those who are affected by the disease. This is why it is essential that countries are in a position to develop and maintain adequate diagnostic capabilities that function effectively even in times of crisis. Analyzing SARS-CoV-2 testing patterns and past shortcomings provides crucial insights for future pandemic preparedness to strengthen health systems and save lives in future health emergencies.

Existing literature has examined SARS-CoV-2 testing rates and capacity in SSA, often attempting to explain low reported morbidity and mortality[15–19].

Studies focused on assessing testing capacities and test introduction[20–22]. A recent mixed-methods review found low testing uptake throughout the pandemic due to both supply-side factors (inadequate service availability, insufficient laboratory capacity) and demand-side factors (poor knowledge, testing apprehension, and political denial)[23].

Interestingly, no study has examined testing uptake relative to actual need, i.e. unmet need for testing. Unmet need occurs when a person with a healthcare need cannot receive adequate services[24]. For COVID-19 testing, unmet need exists when individuals meeting WHO testing criteria (symptomatic or exposed to a confirmed case) do not receive tests, regardless of whether the barrier is supply-side or demand-side[7,25]. Measuring unmet need is more policy-relevant than uptake alone because it captures the gap between population needs and health system response.

This paper examines unmet need for SARS-CoV-2 testing in urban communities in Ghana, Madagascar, and Burkina Faso. We quantify unmet need at the peak of the pandemic and examine socioeconomic inequalities and factors associated with unmet need. Our analysis reveals that over 90% of individuals meeting WHO testing criteria experience unmet need for testing, with the poorest populations facing the greatest barriers while wealthier individuals have significantly better access. These findings underscore the need to strengthen health system preparedness for future public health emergencies.

## Methods

### Study design and setting

Our study makes use of data that were collected from a population-based cross-sectional prevalence survey carried out in urban residential neighborhoods of Ouagadougou and Bobo-Dioulasso (Burkina Faso), Kumasi (Ghana), and Fianarantsoa (Madagascar) from February to May 2021[13]. These urban sites were selected in consultation with local investigators during the pandemic based on the availability of laboratory infrastructure, necessary for serological testing, and safe accessibility to communities for systematic household sampling. Urban settings were prioritized over rural areas because official surveillance systems registered higher case counts in urban centers.

A Joint External Evaluation (JEE) tool was created to "measure country-specific status and progress in developing capacity to prevent, detect and rapidly respond to public health threats"[26,27]. Assessments were made in categories surrounding prevention, detection, and response, with scores given from 1-5 (1: No capacity, 2: Limited capacity, 3: Developed capacity, 4: Demonstrated capacity, 5: Sustainable capacity)[27]. JEE scored Ghana's National Laboratory System at 2.25/5 overall, with laboratory testing for priority diseases at 3/5, but limitations in specimen referral (2/5), point-of-care diagnostics (2/5), and laboratory quality systems (2/5)[27]. Test and treatment centers were established across the country, leveraging existing influenza surveillance laboratory infrastructures[28]. Ghana enhanced diagnostic efficiency through drone delivery of test samples and pooled testing[29].

Burkina Faso had a single laboratory in Bobo-Dioulasso responsible for SARS-CoV-2 testing at the onset of the pandemic[30]. By May 2020, seven facilities existed in these two cities, expanding to 36 facilities nationally by June 2021[31]. The JEE scored Burkina Faso's laboratory system at 3/5 for detection of priority diseases, 3/5 for modern diagnostics, 3/5 for laboratory quality systems, and 2/5 in specimen transfer[32].

Fianarantsoa, is the fourth-largest city in the central highlands of Madagascar. A real-time PCR platform was established in January 2020, with testing expanding from one to 12 regions, though capacity did not exceed 1000 tests per day. A 2017 JEE scored Madagascar's laboratory system at 4/5 for detection of priority diseases, but with critical limitations in specimen transfer (1/5) and laboratory quality systems (1/5)[32].

The country was one of ten countries reporting the highest number of COVID-19 cases in Africa even though total number of laboratory-confirmed cases was comparatively low and a seroprevalence study conducted in Fianarantsoa in 2021 found that exposed individuals in that city alone was twice as high as the official case number for the entire country[13].

### Sampling methodology and data sources

Our study uses data from a population-based cross-sectional prevalence survey conducted in urban residential neighborhoods of Ouagadougou and Bobo-Dioulasso (Burkina Faso), Kumasi (Ghana), and Fianarantsoa (Madagascar) in 2021[13,33]. The survey's primary objective was to assess SARS-CoV-2 seroprevalence among the study populations. This analysis addresses a secondary objective, examining unmet need for testing among the same population.

In the original study, the sample size was calculated to estimate seroprevalence with adequate precision[33]. Assuming a conservative seroprevalence estimate of 50% to enable determination of prevalence ranging from 20% to 90%, a significance level (α) of 5%, and relative precision (d) of ±5%, the effective sample size required was 384 participants per site. To account for the clustered survey design, a design effect of 1.45 was applied (assuming an intracluster correlation coefficient of 0.05 and mean cluster size of 10), yielding an adjusted sample size of 557 participants per site. To account for an anticipated 15% non-response rate among eligible households, the target sample size was further adjusted to 655 households per site[33].

Respondents were selected for inclusion into the survey using a two-stage cluster sampling approach following WHO sero-epidemiological standardization protocols[34]. In the first stage, administrative boundaries and national population statistics were used to define sampling clusters using the probability proportional to population size method, ensuring that selection probability was proportional to cluster population size. In the second stage, geographical coordinates were randomly selected within each cluster as households to be recruited into the study. Field teams navigated to these coordinates using OsmAnd software[35] installed on tablets. When coordinates fell directly on a household, that household was approached for recruitment. When coordinates did not fall on a household, standardized protocols were followed to identify the nearest eligible household[33]. Households were defined as groups of two or more individuals sharing meals and/or sleeping in the same dwelling, excluding residential institutions. Household members were eligible if they were aged ≥10 years, provided informed consent (with parental consent and child assent for minors aged 10-17), and had no health contraindications for blood sample collection[33]. Serological samples and questionnaire data were collected and SARS-CoV-2 seropositivity among the respondents determined using a SARS-CoV-2 IgG ELISA assay[13,36].

A total of 3452 households were screened, of which 3346 consented to participate. One household member was selected per household, stratified by age and sex to match national demographic distributions. One member per household was selected to ensure statistical independence of observations and avoid clustering effects within households. This resulted in 3058 individuals for analysis: 1271 from Burkina Faso, 666 from Ghana, and 1121 from Madagascar. The exact number of observations in our analysis, may differ slightly in each model, given that we opted to work only with observations for which we had complete information, not replacing any missing values.

Data was collected using REDCap Mobile App[37], covering demographics, COVID-19 symptoms (past 2 weeks and past year), history of COVID-19 testing, and exposure to confirmed cases. Data analysis was performed using STATA[38]. Blood samples were collected by trained phlebotomists and SARS-CoV-2 IgG seropositivity determined using a validated ELISA assay with high specificity for populations in malaria-endemic regions[13,36].

### Ethical considerations and data protection

Ethical clearance was obtained by the national Ethical Board Committees of each country (National Ethical Committee, Ouagadougou, Burkina Faso, Ministry of Public Health in Antananarivo, Madagascar, Committee on Human Research, Publication and Ethics in Ghana, as well as by the Ethical Commission of the Ärztekammer Hamburg): Reference German Ethics: 2020-10035-BO and 2020-10035-1-BO, Reference Madagascar Ethics: CERBMIORG0000851, No 175-MSANP/SG/AGMED/CNPV/CERBM, Reference Burkina Faso Ethics: No 2020–137/MS/SG/INSP/CRSN, Reference Ghana Ethics: CHRPE/AP/218/20.

Moreover, the study employed a hierarchical pseudonymized ID system, with Screening IDs linked to GPS coordinates for navigation, Household IDs assigned at consent, and Member IDs assigned at sampling. Linkages between participant identities and study IDs were maintained on paper forms with restricted access and stored separately from electronic data at each study site; all electronic data were maintained at the Bernhard Nocht Institute for Tropical Medicine (BNITM) Hamburg, Germany, in a locked room with password-protected, role-based access controls. Data were entered electronically into REDCap using pseudonymized IDs only; no personally identifying information was included in the electronic database. Data were transferred daily via encrypted SSH protocol to a secure server at the BNITM. Exported datasets for statistical analysis contained only pseudonymized IDs with no possibility of re-identification without access to the separately stored paper consent forms.

### Definitions of unmet need, testing, and independent variables

Table 1 provides a definition of all variables included in the analysis and their measurement. The measure of unmet need, our primary outcome variable,

**Table 1 | Variables, their definitions, and measurement**

| Variable | Definition | Measurement |
|---|---|---|
| Outcome variables | | |
| Ever tested for SARS-CoV-2 (COVID test) | Individuals who had ever received a SARS-CoV-2 test | 0 = No 1 = Yes |
| Unmet need measure 1 | Any seropositive individual who did not receive a test for an acute infection | 0 = Met need 1 = Unmet need |
| Unmet need measure 2 | Any individual who had been exposed to a febrile episode of a family member OR had been in contact with a confirmed case outside of the family but did not receive a test for an acute infection | 0 = Met need 1 = Unmet need |
| Unmet need measure 3 | Any individual who had experienced COVID-19-like symptoms over the past 12 months but did not receive a test for an acute infection | 0 = Met need 1 = Unmet need |
| Exposure variables | | |
| Individual's sex | | 0 = Male 1 = Female |
| Individual's age | | Age in years |
| Underlying chronic health condition | At least one underlying chronic health concern | 0 = No 1 = Yes |
| Daily contacts | | Less than 5 5 to less than 10 10 to less than 50 50 or more |
| Individual's travel exposure | At least one travel outside the city of residence | 0 = No 1 = Yes |
| Household size | | |
| Risk perception | Perception to be at high risk for COVID-19* | 0 = No 1 = Yes |
| Household socio-economic status | | Asset-based wealth measure computed using PCA |
| Country | | 0 = Burkina Faso 1 = Ghana 2 = Madagascar |

*Do you consider yourself to be someone at risk of developing a severe form of Corona virus disease?

deserves further explanation. While there is no widely established framework for measuring unmet need for SARS-CoV-2 testing, we constructed three alternative measures of unmet need aligned with WHO testing criteria and adapted to our population-based survey data. In each of the three cases, we looked at the proportion of individuals having received a test (i.e. swab for a PCR test) versus the total number of individuals in need of such a test but without a test at the time the samples and data were collected. Effectively, we measure the difference between the need for a test and actually having received such a test, but we vary the way in which we operationalize this need concept.

The main differences between the unmet need indicators are the definition of need for testing. The numerator captures individuals in need of testing AND never tested for acute SARS CoV-2 (0 = no; 1 = yes), while the denominator - "all respondents with need"- captures all individuals in need (whether or not they were tested). In option 1, need is defined as being seropositive for SARS-CoV-2. In option 2, need is defined as having been exposed to a person with a febrile episode within the family OR contact to a confirmed case outside of the family. In option 3, need is defined as having experienced probable COVID-19 symptoms over the past 12 months. In each of the three options, the rationale for the definition of need relates to the probability of being in need for a test to confirm or exclude possible infection

with SARS CoV-2. To be exhaustive, we included an additional simpler outcome measure, simply differentiating individuals who ever tested (coded as 1) from individuals who did not (coded as 0).

Most exposure variables are self-explanatory and reflect individual and household factors normally associated with access to care[39]. A few variables deserve further explanation. Our survey also asked individuals to report general measures of risk exposure, specifically i) whether they suffered from any chronic conditions (information not verified clinically), ii) whether they travelled outside their area of residence since the pandemic began, iii) and how many people they interacted with daily. In addition, we included a measure of perceived risk, based on a question asking respondents whether they considered themselves to be at risk of developing a severe form of COVID-19. Socio-economic status was computed using an index based on household living conditions and owned assets. We created dummy variables for house ownership, number of rooms, type of toilet facilities, sources of drinking water, cooking fuel, source of energy for lighting, as well as floor, roof, wall material of the home. In terms of assets, dummy variables (yes=1, no=0) were created for whether the household owned a refrigerator, TV, radio, DVD/VCD player, a landline telephone, at least one mobile phone, a car, motorbike, a bicycle, a desktop or a laptop computer, an LPG stove or an electric stove. Using these variables, we created the wealth index using principal component analysis (PCA) and later classified households into quintiles.

### Analytical approach

To address our research objectives, we conducted our analysis in three stages. First, we relied on descriptive statistics to explore distributions and associations between the outcome of interest and exposure variables. Second, we used concentration indices (CIs) and curves (CCs) to examine the nature of socioeconomic-related inequality in our outcome variables. The third stage involved the use of a logistic regression to understand how various socioeconomic and demographic variables affected the likelihood of SARS-CoV-2 testing and unmet need for testing. Our models included all explanatory variables described earlier. All analyses were conducted using pooled data across the three countries. We also conducted separate analyses for each country and reported the results as supplementary material. The pooled analysis was preferred because of the increased degrees of freedom in relation to the increased observations it provided. However, we did not see a significant difference in the results between pooled and country-specific analyses. The wealth indices were constructed separately before and after data pooling to allow for country and pooled analysis. To account for the two-stage cluster sampling design, we adjusted for clustering at the primary sampling unit (PSU) level in all regression analyses. PSUs were defined as arrondissements in Burkina Faso and Madagascar, and districts in Ghana.

**Concentration Indices and curves.** The concentration indices helped quantify the degree of socioeconomic inequality in the outcome. The concentration index is defined as twice the area between the concentration curve and the line of equality[40]. The generalized concentration index (C)[38], suitable for ratio-scale variables, is expressed as:

$$C = 2/\mu \, \text{Cov}\,(h_i, r_i) \tag{1}$$

Where C denotes the estimated concentration index, Cov denotes covariance, $\mu$ is the mean of the outcome variable of the population, $h_i$ is the outcome of individual $i$, and $r_i$ denotes the fractional rank of individual $i$ in the living standards distribution.

In this study, because our outcome variables are dichotomous, we used the Erreygers corrected CI proposed by Erreygers (2009)[41]. The Erreygers concentration index (E) for a binary (0, 1) variable is expressed as:

$$E = 8 * Cov(h_i, r_i) \tag{2}$$

The value of E ranges between $-1$ and $+1$. When there is no inequality, E is zero. A negative value means that the outcome variable is disproportionately concentrated among households with relatively low

socioeconomic status, whereas a positive value implies concentration among households with relatively high socioeconomic status. The closer the absolute value of E is to 1, the greater the level of inequality. The Es are estimated in STATA (as all other analyses) using the conindex command[38]. We also constructed coefficient plots for the Es to present a visual inspection of the indices across the different outcome measures.

The CCs provide a visual complement to the CIs and allow for an intuitive assessment of the magnitude and direction of inequality. If the CC lies above (below) the 45-degree line (line of equality), the outcome variable of interest is concentrated among poorer (richer) population groups.

**Logistic Regression.** We estimated the following model to predict the association between various socioeconomic and demographic variables that may affect the likelihood of SARS-CoV-2 testing and unmet need for testing:

$$y_i = \beta_0 + \sum_{j=1}^{n} \beta_i x_{ij} + \epsilon_i$$

Where $y_i$ is equal to 1 if the respondent has tested for SARS-CoV-2, or has an unmet need for SARS-CoV-2 testing, and 0 otherwise. $x_j$ are the various socioeconomic and demographic factors, including country fixed effects, that are used to predict the outcome variables. $\epsilon_i$ is the error term. The estimated effects are reported as odds ratios using logistic link functions.

## Results

### Univariate descriptive statistics

Table 2 presents summary statistics on the variables used in the study, pooling information on the sample across the three countries, and for the three countries individually. We report the means and standard deviations. Looking at the statistics for the pooled sample, the majority of respondents were female (54%), aged between 20 and 44 (53%), employed (81%), had between 10 and 50 daily contacts (47%), and did not have any underlying condition (89%). The average household size was approximately 6 and was almost equally distributed across wealth quintiles. Testing uptake was very low, with a sample average of 4.3% reporting having been tested for an acute infection of SARS-CoV-2. This pattern was consistent across all three countries, with the highest level of testing in Ghana at 7.2%. In all the countries, irrespective of the specific measure adopted, more than 90% of the study participants had unmet need for SARS-CoV-2 testing. Approximately 41% of the sample had SARS-CoV-2 antibodies. About 11% of the sample had an underlying condition that could exacerbate the effect of an infection. Approximately 18% of the sample perceived themselves to be at high risk of infection.

### Bivariate descriptive statistics

We present bivariate descriptive statistics from the pooled data showing differences in the main outcome variables across individual characteristics (Table 3). The results show that individuals with underlying conditions, perceived risk of COVID-19 and travel history were more likely to test for SARS-CoV-2 and had less unmet need, irrespective of the measure considered. Women had higher unmet need for SARS-CoV-2 testing than men. The proportion of individuals who tested for SARS-CoV-2 increased with wealth with the highest proportion observed among individuals in the highest wealth quintile. Individuals with more than 10 daily contacts were more likely to test for SARS-CoV-2, though unmet need patterns were inconsistent among this population. Employed individuals were more likely to test and less likely to have unmet need, relative to unemployed individuals.

### Socioeconomic-related inequality analysis

Figure 1 plots the concentration indices for the outcome variables used in the study. The horizontal line represents the line of equality; points above this line suggest the outcome measure is concentrated among richer individuals, while points below indicate concentration among poorer individuals. The absolute values of the concentration indices are generally close to zero, indicating a low level of socioeconomic-related inequality. However, they are all highly statistically significant. The concentration index for

**Table 2 | Univariate descriptive statistics**

| Variable | All countries | | Burkina Faso | | Ghana | | Madagascar | |
|---|---|---|---|---|---|---|---|---|
| | Mean | Std. Dev. | Mean | Std. Dev. | Mean | Std. Dev. | Mean | Std. Dev. |
| SARS-CoV-2 test | 0.043 | 0.202 | 0.041 | 0.198 | 0.072 | 0.258 | 0.028 | 0.164 |
| Unmet need measure 1 | 0.951 | 0.215 | 0.955 | 0.207 | 0.922 | 0.268 | 0.964 | 0.186 |
| Unmet need measure 2 | 0.933 | 0.250 | 0.934 | 0.248 | 0.913 | 0.282 | 0.947 | 0.225 |
| Unmet need measure 3 | 0.953 | 0.212 | 0.952 | 0.214 | 0.934 | 0.249 | 0.969 | 0.174 |
| Wealth quintiles | | | | | | | | |
| 1 | 0.201 | 0.401 | 0.200 | 0.400 | 0.204 | 0.403 | 0.201 | 0.401 |
| 2 | 0.202 | 0.401 | 0.201 | 0.401 | 0.199 | 0.399 | 0.199 | 0.400 |
| 3 | 0.199 | 0.399 | 0.199 | 0.400 | 0.204 | 0.403 | 0.201 | 0.401 |
| 4 | 0.202 | 0.402 | 0.200 | 0.400 | 0.195 | 0.397 | 0.199 | 0.400 |
| 5 | 0.196 | 0.397 | 0.199 | 0.400 | 0.199 | 0.399 | 0.199 | 0.400 |
| Female | 0.541 | 0.498 | 0.522 | 0.500 | 0.611 | 0.488 | 0.457 | 0.499 |
| Underlying condition | 0.119 | 0.324 | 0.099 | 0.298 | 0.205 | 0.404 | 0.089 | 0.285 |
| Daily contacts | | | | | | | | |
| Less than 5 | 0.076 | 0.265 | 0.091 | 0.288 | 0.147 | 0.354 | 0.019 | 0.136 |
| 5 to less than 10 | 0.208 | 0.406 | 0.303 | 0.460 | 0.155 | 0.362 | 0.130 | 0.336 |
| 10 to less than 50 | 0.469 | 0.500 | 0.400 | 0.458 | 0.300 | 0.457 | 0.642 | 0.479 |
| 50 or more | 0.247 | 0.431 | 0.205 | 0.400 | 0.490 | 0.486 | 0.209 | 0.407 |
| Travel | 0.319 | 0.466 | 0.255 | 0.436 | 0.490 | 0.500 | 0.287 | 0.453 |
| Household size | 6.498 | 4.531 | 5.999 | 4.081 | 7.686 | 6.977 | 6.520 | 3.340 |
| Risk perception | 0.181 | 0.385 | 0.245 | 0.431 | 0.248 | 0.432 | 0.074 | 0.262 |
| Age categories | | | | | | | | |
| 10 to 19 | 0.224 | 0.417 | 0.258 | 0.438 | 0.121 | 0.326 | 0.248 | 0.432 |
| 20 to 44 | 0.528 | 0.499 | 0.542 | 0.498 | 0.514 | 0.500 | 0.520 | 0.500 |
| 45 and above | 0.248 | 0.432 | 0.200 | 0.400 | 0.366 | 0.482 | 0.232 | 0.422 |
| Employment status | 0.809 | 0.394 | 0.797 | 0.402 | 0.776 | 0.417 | 0.844 | 0.363 |
| N | 3058 | | 1271 | | 666 | | 1121 | |

testing for acute SARS-CoV-2 infection is positive and statistically significant, suggesting that testing was concentrated among socio-economically better-off individuals. Correspondingly, the concentration indices for the three measures of unmet need for testing are negative and statistically significant, indicating that the unmet need for testing is concentrated among individuals from socio-economically poorer households. Concentration curves visualizing these socioeconomic inequality patterns are presented in Fig. 2, confirming the patterns shown in Fig. 1.

### Logistic regression

Table 4 shows the odds ratio results from the logistic regression, pooling samples from all three countries. The findings show a positive association between wealth and SARS-CoV-2 testing, but a negative association between wealth and unmet need for SARS-CoV-2 testing. The magnitude of these results is consistent across all three measures of unmet need. This suggests that individuals from wealthier households are more likely to test for SARS-COV-2 and, therefore, less likely to experience unmet need for testing. It is worth noting that this association was only statistically significant for individuals in the highest wealth quintile. Compared to individuals who had not travelled outside their city, those who had travelled outside their city of residence were more likely to have tested for SARS-CoV-2 and less likely to have unmet need for testing. Employment status had no statistically significant association (at conventional significance levels) on testing or on unmet need for testing. Furthermore, individuals who perceived themselves to be at high risk of contracting the virus were more likely to have tested for SARS-COV-2. They were also less likely to have unmet need for testing, an association that was consistent across all three indicators of unmet need.

### Discussion

Using population-based cross-sectional data from urban communities in Burkina Faso, Ghana, and Madagascar, we quantified unmet need for SARS-CoV-2 testing during the pandemic peak in 2021 and examined associated socioeconomic inequalities. Our analysis revealed that over 90% of urban residents across all three countries experienced unmet need for SARS-CoV-2 testing, while actual testing uptake remained extremely low, ranging from 3.8% overall to just 5.4% in Ghana. Unmet need exhibited significant socioeconomic inequalities: testing was concentrated among wealthier individuals (particularly the highest wealth quintile), those who had traveled outside their city, and those perceiving themselves at high risk of infection, while women and economically disadvantaged populations experienced disproportionately higher unmet need.

This study makes a unique contribution to the literature by presenting the first analysis of unmet need for SARS-CoV-2 testing in urban settings in three sub-Saharan countries. Unlike prior analyses that measured testing uptake[23], our approach identifies the gap between clinical need —defined by WHO testing criteria for symptomatic individuals or those exposed to confirmed cases— and actual receipt of diagnostic testing. Our findings reveal that across countries, in 2021, at time of data collection and at the peak of the pandemic, unmet need for testing remained alarmingly high, consistently above 90%, independently of the specific indicator measure being adopted.

The three measures capture unmet need from different perspectives. The serological measure identifies all undetected infections from a surveillance standpoint, while the exposure and symptom-based measures specifically capture individuals meeting WHO clinical testing criteria (symptomatic or known exposure) who did not receive tests.

## Table 3 | Bivariate descriptive statistics

| | SARS-CoV-2 test | | | Unmet need measure 1 | | | Unmet need measure 2 | | | Unmet need measure 3 | | |
|---|---|---|---|---|---|---|---|---|---|---|---|---|
| | Yes | No | t | Yes | No | t | Yes | No | t | Yes | No | t |
| Underlying conditions | 0.1846 | 0.1169 | −2.3282 | 0.1177 | 0.2373 | 2.7285 | 0.1431 | 0.1687 | 0.6389 | 0.1474 | 0.2247 | 1.9874 |
| Risk perception | 0.3174 | 0.1748 | −4.0817 | 0.1756 | 0.3214 | 2.7625 | 0.2110 | 0.3333 | 2.5740 | 0.2172 | 0.3793 | 3.5434 |
| Travel | 0.5703 | 0.3069 | −6.2977 | 0.2973 | 0.6140 | 5.0830 | 0.3371 | 0.6049 | 4.9110 | 0.3303 | 0.5517 | 4.2722 |
| Wealth quintile | | | | | | | | | | | | |
| 1 | 0.1160 | 0.2043 | 2.2836 | 0.1793 | 0.1429 | -0.6528 | 0.1665 | 0.0694 | −2.1742 | 0.2062 | 0.1026 | −2.2308 |
| 2 | 0.1071 | 0.2063 | 2.5578 | 0.2093 | 0.1224 | -1.4701 | 0.2364 | 0.0833 | −3.0101 | 0.2186 | 0.0769 | −2.9954 |
| 3 | 0.1428 | 0.2005 | 1.4979 | 0.2228 | 0.1429 | -1.3210 | 0.1990 | 0.1667 | −0.6677 | 0.1932 | 0.1795 | -0.2996 |
| 4 | 0.2232 | 0.2016 | −0.5574 | 0.2141 | 0.2245 | 0.1733 | 0.2019 | 0.2917 | 1.8169 | 0.2019 | 0.2308 | 0.6195 |
| 5 | 0.4107 | 0.1875 | −5.8543 | 0.1745 | 0.3673 | 3.4243 | 0.1961 | 0.3889 | 3.9160 | 0.1802 | 0.4103 | 5.0863 |
| Female | 0.5153 | 0.5414 | 0.5820 | 0.5243 | 0.4576 | -0.9995 | 0.5693 | 0.4940 | −1.3375 | 0.5652 | 0.5393 | −0.4819 |
| Daily contacts | | | | | | | | | | | | |
| Less than 5 | 0.0703 | 0.0758 | 0.2306 | 0.0733 | 0.0877 | 0.4048 | 0.0541 | 0.0732 | 0.7311 | 0.0783 | 0.0575 | −0.7104 |
| 5 to less than 10 | 0.1563 | 0.2110 | 1.4925 | 0.2270 | 0.1754 | -0.9104 | 0.1883 | 0.0976 | -2.0584 | 0.2139 | 0.1609 | 1.1825 |
| 10 to less than 50 | 0.4063 | 0.4724 | 1.4671 | 0.4629 | 0.3860 | -1.1369 | 0.4595 | 0.4390 | -0.3585 | 0.4506 | 0.4713 | 0.3775 |
| 50 or more | 0.3672 | 0.2407 | -3.2538 | 0.2367 | 0.3509 | 1.9639 | 0.2981 | 0.3902 | 1.7514 | 0.2571 | 0.3103 | 1.1057 |
| Age categories | | | | | | | | | | | | |
| 10 to 19 | 0.0233 | 0.2340 | 5.6359 | 0.2567 | 0.0172 | −4.1627 | 0.23965 | 0.0244 | −4.5419 | 0.2209 | 0.02272 | −4.4649 |
| 20 to 44 | 0.6589 | 0.5224 | −3.0436 | 0.4926 | 0.5862 | 1.3911 | 0.5353 | 0.7195 | 3.2492 | 0.5346 | 0.7160 | 3.3401 |
| 45 and above | 0.3178 | 0.2436 | −1.9131 | 0.2507 | 0.3966 | 2.4831 | 0.2250 | 0.2561 | 0.6492 | 0.2444 | 0.2614 | 0.3608 |
| Employed | 0.8425 | 0.8065 | −1.0063 | 0.7908 | 0.8246 | 0.6105 | 0.8280 | 0.9012 | 1.7020 | 0.7996 | 0.8506 | 1.1612 |
| N | 3041 | | | 1211 | | | 1244 | | | 1881 | | |

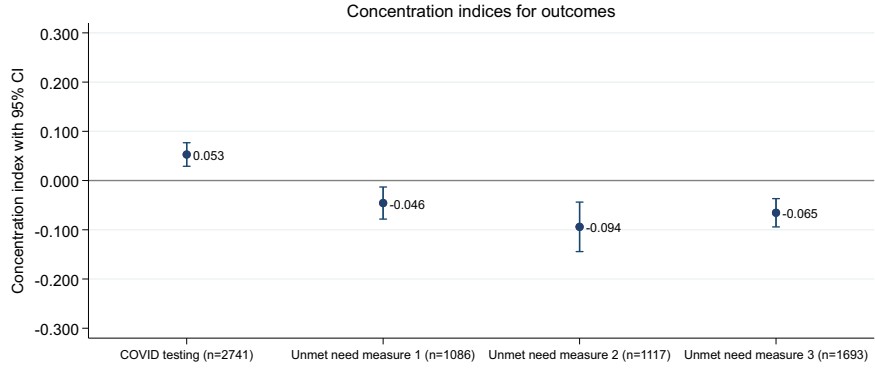

**Fig. 1 | Coefficient plot of concentration indices with 95% Confidence Interval (CI).** Concentration indices quantifying socioeconomic inequality in COVID-19 testing and unmet need (pooled across Ghana, Burkina Faso, and Madagascar). Dots represent concentration index estimates, vertical lines represent 95% confidence intervals. Four outcomes are presented: COVID testing (0.053; CI 0.036; 0.070), serological-based unmet need (measure 1) (−0.046; CI −0.074; −0.017), exposure-based unmet need (measure 2) (−0.094; CI −0.127; −0.061), and symptom-based unmet need (measure 3) (−0.065; CI −0.088; −0.043). Positive values indicate concentration among wealthier individuals (pro-rich inequality), negative values indicate concentration among poorer individuals (pro-poor inequality). Testing is concentrated among wealthier populations, while all unmet need measures are concentrated among poorer populations, demonstrating substantial socioeconomic inequalities in testing access. The standard errors for all estimates presented are cluster-adjusted.

Conversely, the proportion of individuals who reported ever having received a COVID-19 test was extremely low, ranging from below 3% in Madagascar to just above 5% in Ghana. While there are no other studies reporting on unmet need for SARS-CoV-2 testing, our findings are generally consistent with prior studies suggesting low uptake of COVID-19 testing across African countries, ranging from a low of less than 2% reported in Ethiopia in 2020 to a high of nearly 11% reported in Lagos State in Nigeria in 2021[15,16,42]. This consistency leads to the assumption that a similar pattern of unmet need for SARS-CoV-2 testing would probably be detected, if measured, in many other sub-Saharan countries.

This high level of unmet need bears two important policy implications[1]. It provides corroborating evidence that the true impact of the pandemic might have been largely underestimated due to insufficient testing capacity[42], and[2] it highlights important shortcomings in health system preparedness that need to be addressed in preparation for future health emergencies. We note that capacity to test for COVID-19 was built across 42 African countries in just one month, representing a success story for the continent[43]. Nonetheless, as clearly shown by our analysis, the efforts made fell short of meeting real need, because most health systems were not prepared *ex ante* to deal with an emerging health crisis but were put into action to develop a response only once the crisis manifested. In light of this consideration, the inability to meet

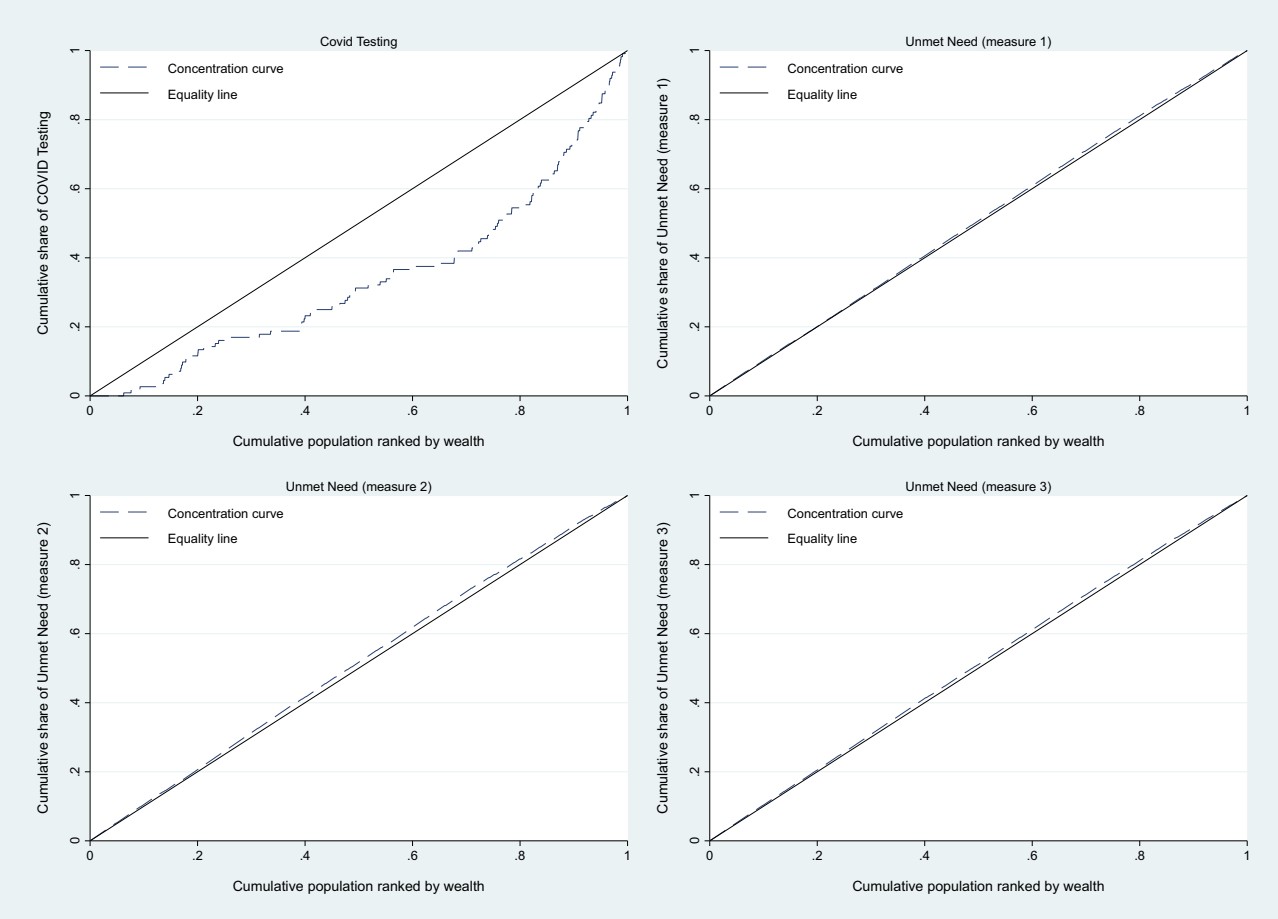

**Fig. 2 | Concentration curves for COVID-19 testing and unmet need.** Concentration curves plot the cumulative distribution of a variable, in our case COVID-19 testing and unmet need for testing, across the population ranked by socioeconomic status. The diagonal line represents perfect equality. Curves below the line of equality indicate a concentration among wealthier individuals (pro-rich distribution), while curves above indicate a concentration among poorer individuals (pro-poor distribution). In our case, the curve for COVID-19 testing is well below the equality line, so strongly concentrated among wealthier populations. The curves lie close to or slightly above the equality line across all three measures, indicating modest inequality towards poorer populations (pro-poor distribution).

testing needs for COVID-19 should serve as an example that steady investments in laboratory and diagnostics infrastructure, including a well-qualified workforce, are needed to build resilient health system preparedness[44]. Moreover, looking forward, health system preparedness can be enhanced by learning from and replicating the experience of countries that implemented expanded testing capacity through innovative strategies. Examples come from Zimbabwe, which relied on task-shifting and decentralization[45], and from Lesotho and Zambia, which relied on community-based case-finding and testing led by community-health workers[46].

An important consideration in interpreting these findings is the extent to which unmet need reflects supply-side constraints versus demand-side barriers. A mixed-methods review identified supply-side capacity as a major structural barrier, while demand-side barriers were often conditional responses to service delivery design[23]. Political leadership also shaped demand, with public willingness to test increasing when leaders modeled testing behavior[23]. Our finding that wealthier individuals and travelers were considerably more likely to test corroborates this: when resources existed to overcome barriers, testing occurred. This underscores that effective health system preparedness requires both dimensions: expanding laboratory infrastructure while ensuring testing services are affordable, accessible, privacy-protected, and supported by strong political leadership and community engagement.

Our findings also indicate considerable socio-economic inequities in the distribution of SARS-CoV-2 testing[47–49]. Mirroring this inequity, our concentration index analysis indicated that unmet need for testing was disproportionately concentrated among the poorest segments of society, a finding which was confirmed by the multivariate regression model. Our findings are neither surprising nor unique to COVID-19, since inequalities in access to diagnostics have long been recognized as a perennial challenge, threatening both population health and global public health security. In turn, inequalities in access to diagnostics often reflect a more generalized problem of accessibility to healthcare services for the poorest segments of society, due to financial and other barriers[50]. Detection of important socio-economic inequalities in unmet need suggests that strategies aimed at enhancing health system preparedness should also consider affordability and financial protection, in line with the core objectives of Sustainable Development Goal 3[51]. Expanding testing capacities is necessary but not sufficient to achieve health system preparedness fully in line with the SDG objective of Leaving No One Behind. Achieving this goal also requires ensuring that people pay no fees for diagnostics at point of use and, if possible, reducing direct and indirect costs associated with travel to a minimum[48]. In Ghana, for instance, voluntary SARS-CoV-2 testing was only available at selected facilities and upon payment of a fee, limiting access to diagnostics for the poor.

Finally, our findings provide some evidence that risk perceptions were positively associated with individuals' decisions to test for SARS-CoV-2 and negatively associated with unmet need for testing. The finding was consistently significant across all measures of uptake and unmet need. In line with this finding, we note that a recent qualitative review of factors associated with SARS-CoV-2 testing in Africa revealed how risk perceptions, defined in relation to fear of contracting the disease and bearing its consequences, could

## Table 4 | Unmet need for SARS-CoV-2 testing in all countries

| | SARS-CoV-2 test | Unmet need measure 1 | Unmet need measure 2 | Unmet need measure 3 |
|---|---|---|---|---|
| Wealth quintile (ref=1) | | | | |
| 2 | 0.853 | 1.381 | 1.001 | 2.000 |
| | [0.355,2.049] | [0.401,4.756] | [0.155,6.483] | [0.424,9.431] |
| 3 | 1.196 | 1.118 | 0.518 | 0.567 |
| | [0.597,2.399] | [0.263,4.749] | [0.106,2.528] | [0.210,1.531] |
| 4 | 1.750 | 1.030 | 0.314 | 0.499 |
| | [0.787,3.889] | [0.383,2.769] | [0.061,1.607] | [0.178,1.400] |
| 5 | 3.387*** | 0.386 | 0.252 | 0.260** |
| | [1.505,7.623] | [0.119,1.255] | [0.042,1.516] | [0.091,0.746] |
| Female | 0.916 | 1.545 | 1.148 | 0.951 |
| | [0.537,1.563] | [0.768,3.107] | [0.626,2.105] | [0.550,1.643] |
| Underlying condition | 1.237 | 0.609 | 1.087 | 0.671 |
| | [0.602,2.542] | [0.182,2.038] | [0.410,2.877] | [0.298,1.512] |
| Travel | 1.881** | 0.380** | 0.539** | 0.576** |
| | [1.106,3.198] | [0.168,0.857] | [0.295,0.983] | [0.338,0.982] |
| Risk perception | 1.811** | 0.379*** | 0.545** | 0.465*** |
| | [1.079,3.040] | [0.186,0.775] | [0.313,0.948] | [0.271,0.800] |
| Daily contacts (ref=less than 5) | | | | |
| 5 to less than 10 | 1.288 | 1.348 | 3.282 | 1.188 |
| | [0.341,4.863] | [0.279,6.501] | [0.730,14.762] | [0.336,4.207] |
| 10 to less than 50 | 1.372 | 1.233 | 1.859 | 0.895 |
| | [0.290,6.484] | [0.141,10.805] | [0.316,10.929] | [0.173,4.632] |
| 50 or more | 2.382 | 0.862 | 1.340 | 0.834 |
| | [0.499,11.362] | [0.090,8.240] | [0.242,7.415] | [0.166,4.179] |
| Household size | 0.967 | 1.071 | 1.114 | 1.040 |
| | [0.921,1.015] | [0.954,1.202] | [0.973,1.276] | [0.972,1.112] |
| Age | 1.010 | 0.983 | 0.992 | 0.998 |
| | [0.998,1.022] | [0.959,1.007] | [0.975,1.009] | [0.982,1.014] |
| Employed | 1.235 | 0.624 | 0.563 | 0.677 |
| | [0.576,2.649] | [0.185,2.103] | [0.237,1.337] | [0.294,1.557] |
| Country (Ref: Burkina Faso) | | | | |
| Ghana | 1.189 | 1.496 | 1.379 | 1.157 |
| | [0.678,2.086] | [0.500,4.475] | [0.469,4.053] | [0.596,2.246] |
| Madagascar | 0.801 | 0.948 | 1.082 | 1.492 |
| | [0.473,1.357] | [0.477,1.884] | [0.585,2.001] | [0.847,2.629] |
| Constant | 0.009*** | 70.625*** | 26.826** | 65.491*** |
| | [0.002,0.049] | [6.797,733.805] | [1.975,364.448] | [10.174,421.584] |
| N | 2101 | 832 | 875 | 1290 |

Exponentiated coefficients.
*$p < 0.10$, **$p < 0.05$, ***$p < 0.01$.

serve both as a motivator to test, out of concern to seek prompt treatment, and as a demotivator to test, out of concern about facing social isolation and, in the worst cases, stigma[23]. Appraising our findings in light of this qualitative evidence points to the importance of investing in adequate communication and educational campaigns to reduce stigma and increase diagnostic acceptability, as essential elements of health system preparedness. This concern, that a diagnosis may be associated with social isolation and stigma, is not unique to COVID-19 and has been widely reported in the literature for other infectious diseases, including TB and HIV[52–54].

The strength of our study lies not only in its innovative conceptualization and operationalization of unmet need as an outcome measure, but also in its reliance on population-based primary data collected across multiple urban centers using rigorous probability sampling.

Several limitations should be acknowledged. First, our sample was drawn exclusively from urban settings, which may have overestimated testing uptake and underestimated unmet need. True unmet need at the country level was likely even greater when accounting for rural areas, where access to healthcare services is typically poorer. At the time of the study, Rapid Diagnostic Tests (RDTs) were not yet available; testing required accessing laboratory facilities, generally located in urban areas. Even after RDTs became available, they continued to be dispensed primarily through the healthcare system, largely perpetuating urban-rural disparities in access[55]. Therefore, our findings reflect urban measures of unmet need, not country-wide estimates.

Second, our measures of unmet need rely on retrospective accounts of testing behavior and cannot trace the exact time when the event occurred,

https://doi.org/10.1038/s43856-026-01637-z **Article**

potentially introducing recall bias. As such, our measures of unmet need reflect the discrepancy between individuals potentially needing a test and actually receiving one, but the timing of need and testing cannot be determined. The limitation indicates that our analysis generates lower-bound estimates of unmet need. For example, seropositive individuals had been exposed to SARS-CoV-2 and might have needed testing multiple times during the study year.

Third, our analysis includes only demand-side factors. Although the seroprevalence survey collected geo-spatial information on households, we could not determine distances to the testing laboratories or facilities. Similarly, since our initial primary objective was to estimate seroprevalence rates and not to focus on health service delivery, our survey contained no information on the quality of the testing facilities available to participants. Therefore, we could not assess heterogeneity in household decisions related to quality of testing. While our analysis of demand-side factors provides insight into how supply may need to change to enhance health system preparedness, future analyses should also account for supply-side factors.

Fourth, our analysis cannot discern to what extent unmet need is driven by demand versus supply-side factors, i.e. whether the mismatch between need and diagnostics is due to insufficient laboratory facilities, people's unwillingness to be tested, or both. Overcoming unmet need would require facilities to be free of charge and easily accessible. Lastly, we note that generalizability of the findings is limited by the relatively small sample size and geographic scope of the sampling strategy in the three countries studied. Our findings reflect urban settings in Burkina Faso, Ghana, and Madagascar during 2021, and patterns of unmet need may differ in other sub-Saharan African contexts. Rural settings likely experienced even higher levels of unmet need given greater distances from testing facilities and more limited healthcare resources. While we argue that the fundamental challenges of insufficient testing capacity and socio-economic inequalities in access are likely widespread across the region, the specific magnitude of unmet need and nature of barriers may vary across countries and settings. We also acknowledge that our complete-case analysis may bias estimates if missing data are systematically related to key variables, such as socioeconomic status or testing history, which influence testing uptake.

## Conclusions
Our study demonstrates that unmet need for SARS-CoV-2 testing represented a critical challenge in three sub-Saharan African countries during the peak of the COVID-19 pandemic, even within urban settings, where laboratory facilities were, at least in theory, established and accessible. Moreover, unmet need was largely concentrated among the poorest segments of the population, showing that inequities in access to diagnostics persisted across countries. This calls for urgent measures to strengthen health system preparedness for both routine care provision and future health crises. Such measures should target increasing diagnostic capacities while also ensuring outreach to vulnerable populations. Lifting financial barriers and increasing the number of healthcare access points to ensure inclusion should become integral elements of health system preparedness.

## Data availability
The complete pseudonymized dataset ($n = 3,058$ individuals, 195 variables) supporting all analyses presented in this manuscript, including the data used to generate Figs. 1 and 2 is provided as Source Data (.csv file) with this submission and will be made publicly available as supplementary material upon publication. Readers can access the source data file through the online version of this article.

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

## Acknowledgements

No specific funding was received for this work. The original data collection was supported by German Ministry of Health (ZMVI1-2520COR001), COVID-19 National Trust Fund Ghana (NTD/COVID-19/19/F/F/006), and Federal Ministry of Education and Research (01KI20210).

## Author contributions

M.D.A., N.S.S., and J.H.A. conceived of the study design and oversaw all phases of the study, from data analysis to writing. J.N. led the analysis with support from M.D.A. and E.L. N.S.S., E.L., T.A.Q., V.B., B.C., A.S., A.A.A.A., D.F., R.A.R., J.M., J.H.A. and Au.S. contributed to the collection of the original data that were used for the study. J.N., M.D.A. and N.S.S. drafted the manuscript with contribution from all authors. All authors have reviewed and approved of the final version of the manuscript.

## Funding

## Competing interests

The authors declare no competing interests.

## Additional information

[1]Centre for Social Policy Studies, University of Ghana, Accra, Ghana. [2]Kumasi Centre for Collaborative Research in Tropical Medicine, Kwame Nkrumah University of Science and Technology, Kumasi, Ghana. [3]Department of Global and International Health, Kwame Nkrumah University of Science and Technology, Kumasi, Ghana. [4]Department of Infectious Disease Epidemiology, Bernhard Nocht Insitute for Tropical Medicine, Hamburg, Germany. [5]German Center for Infection Research (DZIF), Hamburg-Borstel-Lübeck-Riems, Hamburg, Germany. [6]Centre de Recherche en Santé de Nouna, Nouna, Burkina Faso. [7]German Center for Infection Research (DZIF), Heidelberg, Heidelberg, Germany. [8]Heidelberg Institute of Global Health, University Hospital and Medical Faculty, Heidelberg University, Heidelberg, Germany. [9]Department of Molecular Medicine, Kwame Nkrumah University of Science and Technology, Kumasi, Ghana. [10]Research Group: Implementation Research, Bernhard Nocht Institute for Tropical Medicine, Hamburg, Germany. [11]Department of Infectious Diseases, University of Fianarantsoa Andrainjato, Fianarantsoa, Madagascar. [12]Department of Tropical Medicine I, University Medical Center Hamburg-Eppendorf (UKE), Hamburg, Germany. [13]These authors contributed equally: Jacob Novignon, John H. Amuasi. [14]These authors jointly supervised this work Nicole S. Struck, Manuela De Allegri. ✉e-mail: nicole.gilberger@bnitm.de

