## [Transparent Peer Review file · Communications Medicine]

Inequalities and determinants of unmet need for SARS-CoV-2 testing in Ghana, Burkina Faso and Madagascar (2020 – 2021)

Corresponding Author: Dr Nicole Struck

Version 0:

Reviewer comments:

Reviewer #1

(Remarks to the Author)

I am pleased to review this article which sought to estimate the unmet needs on the SARS-Cov-2 test in 3 countries of Sub-Saharan Africa. This study is very important and allows us to understand the challenges related to the response to Covid-19 because screening is an important step in the fight against an epidemic and conditions the management directly.

The article was well written however, here are some comments to improve the quality of this manuscript:

1. Background

The background should be more concise and avoid information that is not directly related to the purpose of this manuscript. In addition, it is not clear what information is already available from these 3 countries at this level and the first description of the situation in these countries appears in the methodology. It is important to orient this section to the objective of the study and we must see by reading the information from these 3 countries that show the gap requiring this study. In addition there are some formal errors concerning the references that are stuck to the words, if you can review this for example the references better add in some places it is stuck even to the word ex reference 12, (7,8), (4), (3), (42)..

In addition, there are statements but which cannot be referenced, for example:

« The COVID-19 pandemic has exposed weaknesses in health systems' preparedness to respond to public health emergencies all over the globe. Shortcomings were observed across all areas of pandemic management, from prevention and detection to reporting and treatment. Gaps were particularly pronounced in low-income settings, where limited capacities and underfunded infrastructure impaired effective response. »

« Evidence from SARS-CoV-2 seroprevalence studies and alternate mortality measurements revealed a considerable discrepancy between officially reported statistics on acute infections (i.e. SARS-CoV-2 PCR results or rapid tests) and the retrospectively collected serological data. This suggests that the true burden of the pandemic was higher than formally reported. »

2. Methodology

I think that the information on the incidence, testing should not be here, by reading the introduction we should see all this and understand why this study is important obviously in these 3 countries. It is not correct that the first information on these countries is included in the methodology. Here the situation can be described in a more subtle way but we should already see something about the 3 countries at the introduction

I suggest talking about Study design and setting: here start with this sentence:

« Our study makes use of the data that were collected from a population-based cross-sectional prevalence survey carried out in urban residential neighborhoods of Ouagadougou and Bobo-Dioulasso (Burkina Faso), Accra, Kumasi, and Tamale (Ghana), and Fianarantsoa (Madagascar) in 2021 »

Describe these regions rather but in a more susceptible way instead of the description made of the 3 countries. Add the sample size calculation and the parameters used for screening 2, 540 households. Even though the sampling technique has been described in another paper, one should see here how it was done and not refer readers to read it in the other paper. In all, specify the software that was used to analyze the data.

In the ethical considerations, add what you had done to respect the ethical principles, for example how you had ensured anonymity if it was done, was the consent administered? the principle of beneficence how it was respected. All this must be described.

3. Results

No word on the socio-demographic characteristics of the participants... This is the first step even before unmet need.

4. Discussion

In the first paragraph, stay on your main results and then it would be more logical to read how you discussed them. I don't think that from the beginning you are talking about your strengths, make sure to leave that down where you talk about strengths and limitations. There are a lot of rumors about Covid in SSA, saying that the disease was more in Europe, do you really think there was an unmet need for testing? Even if we recognize the limited capacity of countries to test cases, but if it was available the population would have come? Please take this into account also in your discussion and write your conexte maybe the situation would be different in your country.

Good luck!!!!

Reviewer #2

(Remarks to the Author)

Reviewer Comments

1. May you add the timeframe in the title – usually helps for context
2. Introduction: I think additional context is needed – there are supply-side factors as well as demand-side factors in unmet need. Consider acknowledging this and then show how your methods address this.
3. In line 34 (abstract) – please include timeframe when the data were collected
4. Line 36 of abstract – consider naming the three different measures you used
5. Line 272: consider listing some of the variables you controlled for, and why you chose them.
6. Line 274: you indicate that pooled analyses were preferred because it reduces degrees of freedom. It is the opposite, increasing the sample increases degrees of freedom.
7. Line 277: does this imply that the countries were relatively homogeneous? And what about the tightness of the confidence intervals given smaller sample sizes?
8. While the logistic regression is good, may you consider doing a decomposition analysis (Oaxaca-Blinder), so that the unmet need can be broken into explained components (education, wealth, location) and unexplained (which can show the structural issues that need to be addressed)
9. Line 379: see previous comment on demand versus supply-side factors in unmet demand
10. Line 466: but given the dearth in testing points early in the pandemic, some geospatial analysis can still be conducted.
11. Line 466: I don't think your analysis could have told a part supply side from demand side factors. Please rewrite.
12. Line 470: see comments on Oaxaca-Blinder decomposition above
13. Line 794: While these coefficient plots give information, may you consider showing concentration curves instead? That will show the nature of inequality better – for example, how worse off the poor are compared to the rich

Version 1:

Reviewer comments:

Reviewer #1

(Remarks to the Author)

At this stage, i think that all the comments that i made have been addressed

Reviewer #2

(Remarks to the Author)

Reviewer 2 – Rereview

Thank you for incorporating my initial suggestions. I have a series of mostly editorial comments.

1. Because the work revolved around urban areas, there are generalizability concerns that you acknowledge. Please insert “urban” in the abstract on line 36 of the updated PDF. It should read, “We used urban population-based....”
2. From the preceding point, references to the population should be framed as urban residents
3. Line 194: I missed this before – the selection of one member per household. How do you address clustering within the cluster/PSU level? Is this not the main design feature of the original data collection? I am not sure how you'd handle this in the context of the concentration indices – perhaps apply survey weights before this. It's easy to handle this for logistic regressions.
4. Line 237: This is quite confusing – shouldn't the numerator change with the changing definitions of the denominator? For example, if the denominator is exposure within a family, why should the numerator be all people in need for testing and never tested? Shouldn't the numerator be all exposed people in need of testing and never tested?
5. On seropositivity – does this not indicate past infection, not necessarily clinical indication of testing at the time.
6. It would have been great to provide country-disaggregated concentration curves. Were the wealth indices created before or after pooling? Please add a line on this.
7. I missed that Burkina Faso had two cities in the analysis while the other countries only had one city. No need to do anything about this, but for future analyses, consider adding site fixed effects.
8. Add a sentence in the limitations section on how complete case analysis can bias estimates if missingness is related to

some key variables like SES or testing

9. Please recheck duplicated references – 32 and 33 for example

Version 2:

Reviewer comments:

Reviewer #2

(Remarks to the Author)

Many thanks for giving me a chance to review this manuscript and for incorporating my and other reviewer suggestions. I have no further comments.

Point-by-point response to the reviewers

We are grateful to the reviewers for this overall positive review of our work. Please note that we have made extensive revisions throughout the manuscript to address all comments thoroughly, which has resulted in substantial track changes in the marked version.

Reviewer #1 (Remarks to the Author):

I am pleased to review this article which sought to estimate the unmet needs on the SARS-Cov-2 test in 3 countries of Sub-Saharan Africa. This study is very important and allows us to understand the challenges related to the response to Covid-19 because screening is an important step in the fight against an epidemic and conditions the management directly.

The article was well written however, here are some comments to improve the quality of this manuscript:

Background

The background should be more concise and avoid information that is not directly related to the purpose of this manuscript. In addition, it is not clear what information is already available from these 3 countries at this level and the first description of the situation in these countries appears in the methodology. It is important to orient this section to the objective of the study and we must see by reading the information from these 3 countries that show the gap requiring this study. In addition there are some formal errors concerning the references that are stuck to the words, if you can review this for example the references better add in some places it is stuck even to the word ex reference 12, (7,8), (4), (3), (42).. In addition, there are statements but which cannot be referenced, for example:

« The COVID-19 pandemic has exposed weaknesses in health systems' preparedness to respond to public health emergencies all over the globe. Shortcomings were observed across all areas of pandemic management, from prevention and detection to reporting and treatment. Gaps were particularly pronounced in low-income settings, where limited capacities and underfunded infrastructure impaired effective response. “

« Evidence from SARS-CoV-2 seroprevalence studies and alternate mortality measurements revealed a considerable discrepancy between officially reported statistics on acute infections (i.e. SARS-CoV-2 PCR results or rapid tests) and the retrospectively collected serological data. This suggests that the true burden of the pandemic was higher than formally reported. “

Authors' response: We thank the reviewer for this important feedback. We have substantially revised the introduction to address all points raised:

- As requested, the introduction is now more concise and focused. We have streamlined content to avoid information not directly related to the purpose of this manuscript.
- We added available information from Burkina Faso, Ghana, and Madagascar to the introduction, including the epidemiological context that was previously in the methods section (see lines 87-95 in the introduction of the marked version).
- We have corrected all spacing errors between text and citations throughout the manuscript, including references 3, 4, 7, 8, 12, and 42 as noted by the reviewer.
- We apologize for the unreferenced statements flagged by the reviewer. We have now added appropriate citations for all previously unreferenced claims: Marked version line 57, “The Covid-19 pandemic exposed weaknesses in health system preparedness”: Kandel et al. 2020. Marked version line 60, “Gaps were particularly pronounced in low-income settings”: Agwanda et al. 2021. Marked version line 80, Seroprevalence studies in South Sudan and Zambia showing underreporting: Wiens et al. 2021;

Mulenga et al. 2021. Marked version lines 87 – 95, Evidence from Ghana, Burkina Faso, and Madagascar: Struck et al. 2022; Amuasi et al. 2025.

2. Methodology

I think that the information on the incidence, testing should not be here, by reading the introduction we should see all this and understand why this study is important obviously in these 3 countries. It is not correct that the first information on these countries is included in the methodology. Here the situation can be described in a more subtle way but we should already see something about the 3 countries at the introduction

I suggest talking about Study design and setting: here start with this sentence:

« Our study makes use of the data that were collected from a population-based cross-sectional prevalence survey carried out in urban residential neighborhoods of Ouagadougou and Bobo-Dioulasso (Burkina Faso), Accra, Kumasi, and Tamale (Ghana), and Fianarantsoa (Madagascar) in 2021”

Describe these regions rather but in a more susceptible way instead of the description made of the 3 countries.

Authors' response: We have moved all epidemiological information about Burkina Faso, Ghana, and Madagascar from the methods section to the introduction (marked version lines 87-95) and we started the section with the sentence “Our study makes use of the data” as suggested by the reviewer (marked version lines 145 – 148). We further aimed to provide context on the study countries without extensive detail.

Add the sample size calculation and the parameters used for screening 2, 540 households. Even though the sampling technique has been described in another paper, one should see here how it was done and not refer readers to read it in the other paper.

Authors' response: We have added more details on sample size calculation (marked version lines 265-272) and sampling methodology (marked version lines 293-307) to Section 2.2 (Sampling Methodology and Data Sources).

In all, specify the software that was used to analyze the data.

Authors' response: We have added specification of the software used for data collection (marked version line 303) and data analysis (line 305).

In the ethical considerations, add what you had done to respect the ethical principles, for example how you had ensured anonymity if it was done, was the consent administered? the principle of beneficence how it was respected. All this must be described.

Authors' response: We have expanded the description of data protection measures in Section 2.3 (Ethical Considerations and Data Protection, marked version lines 318 – 328).

3. Results

No word on the socio-demographic characteristics of the participants... This is the first step even before unmet need.

Authors' response: We thank the reviewer for this important observation. We have restructured the results section to begin with participants' socio-demographic characteristics (marked version lines 434 – 437).

4. Discussion

In the first paragraph, stay on your main results and then it would be more logical to read how you discussed them. I don't think that from the beginning you are talking about your strengths, make sure to leave that down where you talk about strengths and limitations.

Authors' response: We thank the reviewer for this suggestion, which has improved the discussion structure. The first paragraph (lines 492-500) now focuses on presenting the main findings, while methodological strengths and limitations (previously lines 506 - 508 and 513 – 521) have been moved to the Methodological Considerations section (lines 593-637).

There are a lot of rumors about Covid in SSA, saying that the disease was more in Europe, do you really think there was an unmet need for testing? Even if we recognize the limited capacity of countries to test cases, but if it was available the population would have come?

Authors' response: We are grateful for this important comment. The reviewer is correct that perceptions about COVID-19 in sub-Saharan Africa, including beliefs that it was primarily a 'European disease' and concerns about stigma, likely influenced testing behaviour. However, we wanted to clarify what our analysis measures. Our study identifies individuals who met WHO testing criteria (symptomatic or exposed) but did not receive tests, which constitutes unmet need regardless of whether barriers were supply-side or demand-side. Both represent health system preparedness failures that require addressing. While we acknowledge that we cannot disentangle demand from supply-side elements, Reference 23 (Phiri MM, Dunkley Y, Di Giacomo E et al. [2025] Factors influencing uptake of COVID-19 diagnostics in Sub-Saharan Africa: a rapid scoping review. <https://doi.org/10.1371/journal.pone.0305512>) provides important context. The study identifies supply-side capacity as a major structural barrier, explicitly stating “Countries were unable to increase screening and testing due to inadequate laboratory and diagnostic equipment” (Reference 23, Discussion, page 13). Importantly, Phiri et al. also demonstrate that demand-side barriers were often conditional responses to service delivery design rather than fundamental rejection of testing. This suggests that much of the apparent 'unwillingness' was a rational response to poorly designed, inaccessible services that could be overcome through better service delivery models and effective communication campaigns. We have added a new paragraph to the discussion (see lines 544 – 553 in the marked version).

Please take this into account also in your discussion and write your context maybe the situation would be different in your country.

Authors' response: We agree with the reviewer. We have expanded the Discussion to acknowledge the contextual specificity of our findings (lines 631 - 637).

Good luck!!!!

Thank you very much.

Reviewer #2 (Remarks to the Author):

Reviewer Comments

1. May you add the timeframe in the title – usually helps for context

Authors' response: We have amended the title to include the timeframe 2020-2021. This broader timeframe reflects that while data collection occurred in 2021, the survey included retrospective recall of symptoms and testing behavior from the previous 12 months, effectively capturing experiences from 2020 onwards.

2. Introduction: I think additional context is needed – there are supply-side factors as well as demand-side factors in unmet need. Consider acknowledging this and then show how your methods address this.

Authors' response: We thank Reviewer 2 for this important methodological observation regarding the dual nature of unmet need. We have thoroughly revised the introduction to address supply- and demand-side dimensions of unmet need (lines 115 – 121 in the marked version). Regarding methods, our analysis includes variables associated with testing uptake, such as wealth, travel history, perceived risk, and contact patterns, that may relate to both dimensions. We recognize that these variables do not allow us to definitively isolate supply-side from demand-side effects, as they are often interrelated. For example, wealth may reflect both, the ability to afford testing (supply-side) and health literacy (demand-side). We discuss these factors in the discussion (lines 544 – 553 in the marked version) and note that our study cannot quantify their relative contributions in the Methodological Considerations section (lines 624 - 637 in the marked version).

3. In line 34 (abstract) – please include timeframe when the data were collected

Authors' response: We included the time frame of data collection in the abstract (line 35 – 36 in the marked version).

4. Line 36 of abstract – consider naming the three different measures you used

Authors' response: We have added brief descriptors of the three measures (serological-, exposure-, and symptom-based) in the abstract (line 36 - 37). The word limit prevents a more detailed explanation.

5. Line 272: consider listing some of the variables you controlled for, and why you chose them.

Authors' response: We appreciate the reviewer raising this point. All variables included in our analysis are defined in Table 1 and described in Section 2.3 'Definitions of unmet need, testing, and independent variables' (lines 340 - 374). We have added one sentence in Section 2.4 'Analytical approach' (lines 382-383) to explicitly state that regression models included all variables described in Section 2.3. Our selection of variables was limited to the data collected in our original study; therefore, we could include additional relevant factors, such as distance to testing facilities or quality of care, because this information was not collected at that time. This limitation is acknowledged in the Methodological Considerations section (lines 615 - 623).

6. Line 274: you indicate that pooled analyses were preferred because it reduces degrees of freedom. It is the opposite, increasing the sample increases degrees of freedom.

Authors' response: We thank the reviewer for catching this error. This was a typo. We have corrected it (line 385 in marked version).

7. Line 277: does this imply that the countries were relatively homogeneous? And what about the tightness of the confidence intervals given smaller sample sizes?

Authors' response: As the reviewer notes, confidence intervals are larger in country-specific analyses due to smaller sample sizes. However, effect estimates remain stable across all three countries. We cannot distinguish whether this reflects genuine homogeneity among countries or simply consistent effects across contexts.

8. While the logistic regression is good, may you consider doing a decomposition analysis (Oaxaca-Blinder), so that the unmet need can be broken into explained components (education, wealth, location) and unexplained (which can show the structural issues that need to be addressed)

Authors' response: We thank the reviewer for this methodological suggestion. While Oaxaca-Blinder decomposition would provide valuable insights into the relative contributions of specific factors to socioeconomic inequalities, our analysis was designed to identify correlates of unmet need across the population rather than to decompose differences between specific groups.

Our concentration index analysis quantifies socioeconomic inequality (Figures 1 and new Figure 2), and our multivariable regression identifies independent associations between factors (wealth, education, travel, risk perception) and unmet need. Together, these analyses demonstrate both the magnitude of inequality and which factors are associated with unmet need.

We recognize the value of this analytical approach for understanding mechanisms underlying inequality, but we believe it is beyond the scope of the current revision and would be more appropriate as a separate, complementary analysis in future work.

9. Line 379: see previous comment on demand versus supply-side factors in unmet demand

Authors' response: We are grateful for this comment and have added a sentence in the Methodological Considerations section to acknowledge how we are not in the position to discern the two (line 624 - 627).

10. Line 466: but given the dearth in testing points early in the pandemic, some geospatial analysis can still be conducted.

Authors' response: We appreciate this suggestion. While our survey collected GPS coordinates for households, we do not have geospatial data on testing facility locations during the study period. Without information on where testing sites were located, we cannot calculate distances or conduct meaningful geospatial analysis of accessibility. This limitation is noted in our Methodological Considerations section (line 615 - 621).

11. Line 466: I don't think your analysis could have told a part supply side from demand side factors. Please rewrite.

Authors' response: We agree with the reviewer and have revised the text throughout the manuscript to clarify the complexity of distinguishing supply-side from demand-side effects. We cite evidence from a review (Introduction, lines 115-121) showing that unmet need arises from both supply-side and demand-side barriers, discuss the complexity this creates (Discussion, lines 544-553), and explicitly state that our analysis cannot disentangle these effects (Methodological Considerations, lines 624-627).

12. Line 470: see comments on Oaxaca-Blinder decomposition above

Authors' response: As discussed above, implementing Oaxaca-Blinder decomposition would entail focusing on between-group comparisons rather than population-level associations, which we believe would alter the scope of this manuscript and is therefore beyond the scope of this revision.

13. Line 794: While these coefficient plots give information, may you consider showing concentration curves instead? That will show the nature of inequality better – for example, how worse off the poor are compared to the rich

Authors' response: In line with the reviewer's comment, we have now included concentration curves (Figure 2) to complement the concentration indices shown in the coefficient plots (Figure 1).

Point-by-point response to the reviewers

Reviewer 2 – Re-review

Thank you for incorporating my initial suggestions. I have a series of mostly editorial comments.

1. Because the work revolved around urban areas, there are generalizability concerns that you acknowledge. Please insert “urban” in the abstract on line 36 of the updated PDF. It should read, “We used urban population-based...”

Authors reply: We have added the term "urban" to the abstract (line 36) to clarify that our study population was drawn from urban communities in Ghana, Burkina Faso, and Madagascar.

2. From the preceding point, references to the population should be framed as urban residents.

Authors reply: We have reviewed the manuscript to ensure all population references are appropriately qualified as urban. As suggested, we have revised the abstract (line 36) and the plain language summary (line 54). Throughout the manuscript, we use contextually appropriate urban qualifiers: 'urban residential neighborhoods' (lines 133, 167), 'urban sites' (line 135), 'urban settings' (lines 137, 394, 473), and 'urban communities' (line 384). We believe this variation improves readability while maintaining accuracy about the urban context of our findings.

3. Line 194: I missed this before – the selection of one member per household. How do you address clustering within the cluster/PSU level? Is this not the main design feature of the original data collection? I am not sure how you’d handle this in the context of the concentration indices – perhaps apply survey weights before this. It’s easy to handle this for logistic regressions.

Authors reply:

We thank the reviewer for identifying this important issue. We have now re-estimated the logistic regression and concentration indices, accounting for clustering at the PSU level. Fortunately, the “conindex” command in Stata allows for clustering, so this was resolved. The results showed minimal changes, with only two variables showing minor changes in statistical significance. All main findings remain consistent. We included an updated Table 4 and Figure 1, and added details to the methods section (lines 280 – 282).

4. Line 237: This is quite confusing – shouldn’t the numerator change with the changing definitions of the denominator? For example, if the denominator is exposure within a family, why should the numerator be all people in need for testing and never tested? Shouldn’t the numerator be all exposed people in need of testing and never tested?

Authors reply: Many thanks to the reviewer for this important observation. We have now revised the statement to show that the main difference between the unmet need measures in relation to the definition of need. The numerator does not remain constant across the different measures. We have made this clearer.

5. On seropositivity – does this not indicate past infection, not necessarily clinical indication of testing at the time.

Authors reply: The reviewer makes a valid methodological point and we agree with this observation. Seropositivity indicates past infection but does not confirm individuals met WHO testing criteria at the time. We have clarified in the Discussion (lines 401 - 404) that the serological measure captures all undetected infections (surveillance perspective), while the symptom and exposure-based measures specifically capture WHO criteria-based unmet need (clinical perspective). The consistency across all measures (>90%) demonstrates our findings are robust regardless of this conceptual difference.

6. It would have been great to provide country-disaggregated concentration curves. Were the wealth indices created before or after pooling? Please add a line on this.

Authors reply: We have now included country-specific concentration curves and added a line to indicate that wealth indices were constructed before pooling the data to enable the construction of the country CCs, while for the pooled data (and CC), a separate wealth index was constructed after pooling.

7. I missed that Burkina Faso had two cities in the analysis while the other countries only had one city. No need to do anything about this, but for future analyses, consider adding site fixed effects.

Authors reply: We did include country fixed effects, but not site. We appreciate and take note of the reviewer's recommendation for future analysis.

8. Add a sentence in the limitations section on how complete case analysis can bias estimates if missingness is related to some key variables like SES or testing

Authors reply: We are grateful for this suggestion and have added a sentence to reflect on this limitation (lines 507 - 509).

9. Please recheck duplicated references – 32 and 33 for example.

Authors reply: Thank you for this observation. We confirm these are distinct references: one is the WHO Joint External Evaluation for Burkina Faso (2017), the other for Madagascar (2017). Both are IHR capacity assessments but for different countries included in our study.